# Synthesis of FeOOH-Loaded Aminated Polyacrylonitrile Fiber for Simultaneous Removal of Phenylphosphonic Acid and Phosphate from Aqueous Solution

**DOI:** 10.3390/polym15081918

**Published:** 2023-04-17

**Authors:** Rui Zhou, Wusong Xu, Peisen Liu, Shangyuan Zhao, Gang Xu, Qizhong Xiong, Weifeng Zhang, Chaochun Zhang, Xinxin Ye

**Affiliations:** 1Anhui Province Key Lab of Farmland Ecological Conservation and Pollution Prevention, Engineering and Technology Research Center of Intelligent Manufacture and Efficient Utilization of Green Phosphorus Fertilizer of Anhui Province, College of Resources and Environment, Anhui Agricultural University, Hefei 230036, China; 2Key Laboratory of JiangHuai Arable Land Resources Protection and Eco-Restoration, Ministry of Natural Resources, College of Resources and Environment, Anhui Agricultural University, Hefei 230036, China

**Keywords:** polyacrylonitrile fiber, FeOOH, activate peroxydisulfate, organic phosphorus, adsorption

## Abstract

Phosphorus is one of the important metabolic elements for living organisms, but excess phosphorus in water can lead to eutrophication. At present, the removal of phosphorus in water bodies mainly focuses on inorganic phosphorus, while there is still a lack of research on the removal of organic phosphorus (OP). Therefore, the degradation of OP and synchronous recovery of the produced inorganic phosphorus has important significance for the reuse of OP resources and the prevention of water eutrophication. Herein, a novel FeOOH-loaded aminated polyacrylonitrile fiber (PAN_A_F-FeOOH) was constructed to enhance the removal of OP and phosphate. Taking phenylphosphonic acid (PPOA) as an example, the results indicated that modification of the aminated fiber was beneficial to FeOOH fixation, and the PAN_A_F-FeOOH prepared with 0.3 mol L^−1^ Fe(OH)_3_ colloid had the best performance for OP degradation. The PAN_A_F-FeOOH efficiently activated peroxydisulfate (PDS) for the degradation of PPOA with a removal efficiency of 99%. Moreover, the PAN_A_F-FeOOH maintained high removal capacity for OP over five cycles as well as strong anti-interference in a coexisting ion system. In addition, the removal mechanism of PPOA by the PAN_A_F-FeOOH was mainly attributed to the enrichment effect of PPOA adsorption on the fiber surface’s special microenvironment, which was more conducive to contact with SO_4_•^−^ and •OH generated by PDS activation. Furthermore, the PAN_A_F-FeOOH prepared with 0.2 mol L^−1^ Fe(OH)_3_ colloid possessed excellent phosphate removal capacity with a maximal adsorption quantity of 9.92 mg P g^−1^. The adsorption kinetics and isotherms of the PAN_A_F-FeOOH for phosphate were best depicted by pseudo-quadratic kinetics and a Langmuir isotherm model, showing a monolayer chemisorption procedure. Additionally, the phosphate removal mechanism was mainly due to the strong binding force of iron and the electrostatic force of protonated amine on the PAN_A_F-FeOOH. In conclusion, this study provides evidence for PAN_A_F-FeOOH as a potential material for the degradation of OP and simultaneous recovery of phosphate.

## 1. Introduction

Phosphorus (P) is one of the most important elements in the world. It exists in the cells of living organisms, maintains the growth of bones, and joins in almost all physiological and chemical conversions [1]. However, P emissions continue to increase with increasing industrial production and agricultural activities [2]. High concentrations of P (10 mg L^−1^) cause water eutrophication, leading to the excessive growth of algae and aquatic plants, reducing dissolved oxygen levels in the water, and even threatening human safety [3]. Currently, there are still many waters in China that are eutrophic; for example, Chaohu Lake shows an average total phosphorus concentration of 0.215 mg L^−1^ [4]. Moreover, about 65% of European Atlantic coastal waters are eutrophic [5]. Furthermore, P resources are not renewable, and the worldwide phosphorus crisis has gradually attracted people’s attention. Therefore, the removal and recovery of P from wastewater have important research significance.

At present, the recovery and utilization of P in wastewater mainly focus on inorganic phosphorus; however, few studies have reported on organic phosphorus (OP) resource recovery and utilization [6]. OP is one of the important forms of phosphorus in water and livestock wastewater is the major pollution source, accounting for about 10–65% of total phosphorus [7,8]. Furthermore, OP compounds are widely applied as pesticides, scale inhibitors, flame retardants, and plasticizers [9,10]. Ester organophosphorus is the main existing form of OP, which is often detected in water bodies and seriously threatens water safety and human health [11,12]. Hence, the recovery and recycling of OP from wastewater are of imminent importance. However, the direct recovery of organophosphorus in wastewater is difficult. Transforming OP into inorganic phosphorus is an effective way to achieve its recovery and utilization, which has become the focus of people’s attention.

Advanced oxidation processes (AOPs) are regarded as being the most valuable technologies for persistent organic pollutant degradation [13,14,15]. Among them, SO_4_•^−^ is more conducive to the degradation of organic pollutants due to its higher selectivity, longer life cycle, and wider pH adaptability [16,17]. However, the generation rate of SO_4_•^−^ from peroxydisulfate (PDS) or peroxymonosulfate (PMS) is relatively slow. Hence, various methods, including light, heat, bases, and transition metals [18,19], have been successfully used to activate persulfate. Among them, iron-based materials are widely selected due to their advantages of low cost, convenient source, and high catalytic activity. Recently, Fe^0^ [20], FeS [21], Fe_2_O_3_ [22], and FeOOH [23] have been used as potential catalysts for persulfate activation. Among them, FeOOH has been widely used in the water treatment field [23,24,25]. For example, Zhao et al. [23] prepared a ball-milled α-FeOOH/biochar composite to catalyze persulfate degradation of phenol with a high efficiency of 100%. Although important progress has been made in the degradation of organic pollutants in water by activating persulfate with FeOOH, FeOOH is difficult to recycle and easily causes secondary pollution. Different supports, such as silica gel, carbon material, MOF, etc., have been used to load FeOOH for solving the above problems with significant research progress [25,26,27]. However, some of these materials usually have deficits such as low modification degree, harsh preparation conditions, or high cost, which would restrict their actual application. As we all know, suitable support is a prerequisite for improving catalytic activity and stability [28]. Therefore, it is of great significance to exploit new support materials for the construction of a FeOOH-supported catalyst used to activate persulfate for catalytic degradation of OP.

Among all of the potential catalyst supports, fibers have attracted much attention due to their high stability, large specific surface area, prominent recyclability, and ease of surface modification and regulation [29]. Polyacrylonitrile fiber (PANF) is widely used in the textile industry, architecture, and people’s daily lives. Furthermore, PANF is rich in cyano and ester groups, which can be easily converted into various functional groups [30]. Our previous work successfully constructed different functionalized PANFs for organic catalysis [31,32], heavy metal removal [33,34] and phosphate purification [35,36,37]. It was found that a specific microenvironment was formed between the functional moieties and polymer segments, the cooperative effect of which could change the interaction strength, thus accelerating the catalytic or adsorption performance. Nevertheless, using functionalized PANF for the degradation and recovery OP has rarely been reported. In addition, FeOOH has the characteristics of simultaneously activating persulfate to degrade organic pollutants and adsorbing inorganic phosphorus, hence it is theoretically feasible and of great significance to prepare PANF-supported FeOOH for simultaneous OP degradation and inorganic phosphorus recovery.

In this work, a novel polyacrylonitrile fiber loaded with FeOOH (PAN_A_F-FeOOH) was synthesized using the sol-gel method to activate PDS for PPOA degradation. Phenylphosphonic acid (PPOA), a typical OP compound used as an important industrial chemical, was selected as the model pollutant [11]. The PPOA degradation and phosphate adsorption abilities of PAN_A_F-FeOOH were evaluated, and the main factors affecting the degradation of PPOA and phosphate removal were studied. Furthermore, the degradation of PPOA and phosphate purification mechanisms by PAN_A_F-FeOOH are discussed. The stability and feasibility in practical application of the fiber were also evaluated by continuous flow tests and repeatability experiments.

## 2. Materials and Methods

### 2.1. Materials and Instruments

The polyacrylonitrile fiber (PANF, containing about 93% acrylonitrile) and the polyethylene imine (PEI, Mw = 10,000) were purchased from Fushun Petrochemical Co., Ltd.(Fushan, China) and Shanghai Dibo Biotechnology Co., Ltd. (Shanghai, China), respectively. Phenylphosphonic acid (98%) and ferric nitrate (Fe(NO_3_)_3_·9H_2_O) were obtained from Shanghai Aladdin Biochemical Technology Co., Ltd (Shanghai, China). and Shanghai Maclean Biochemical Co., Ltd (Shanghai, China), respectively. Additional reagents used in this study are detailed in Appendix A. The characterization methods for the materials prepared in this experiment are described in Appendix A.

### 2.2. Preparation of FeOOH-Supported Polyacrylonitrile Fiber (PAN_A_F-FeOOH)

PAN_A_F-FeOOH was synthesized mainly through a two-step process, as shown in Figure 1.

Step 1: PANF (1.0 g) and PEI (5.0 g) were dissolved in a reaction kettle containing 20 mL of H_2_O and then transferred to a reaction kettle at 140 °C for 6 h. After the reaction, the fiber was removed with tweezers and washed with H_2_O (70–80 °C) several times until the pH was neutral. The washed fiber was dried overnight in an oven at 60 °C to obtain aminated PAN_A_F with a weight gain of about 38% (relative to PANF). In addition, PAN_A_Fs with different weight gains (3.58%, 7.99%, 15.09%, 20.99%, 24.39%, 28.39%, and 33.39%) were prepared by adding the substrate of PEI with different contents (0.5, 1, 1.5, 2, 2.5, 3, and 4 g), respectively.

Step 2: First, Fe(OH)_3_ colloid solution was prepared: Fe(NO_3_)_3_·9H_2_O (4.04 g) was dissolved in 10 mL of H_2_O, and then it was dropped into 40 mL of boiling deionized water using a drip funnel and stirred vigorously. After cooling at room temperature (25 °C), Fe(OH)_3_ colloid solution was obtained. Then, dried 100 mg PAN_A_F was weighed and added to the colloidal solution of Fe(OH)_3_, and the mixture was stirred for 1 h. Finally, the fiber was filtered and placed in an oven at 105 °C for 6 h to obtain PAN_A_F-FeOOH.

### 2.3. Batch Degradation Test of PPOA

Batch degradation tests were carried out in 100 mL glass bottles at 25 °C. The pH of 50 mL of mixed solution containing 5 mg L^−1^ PPOA and 0.5 mmol L^−1^ PDS was tuned to the target value with 0.1 mol L^−1^ H_2_SO_4_ and 0.1 mol L^−1^ NaOH. Subsequently, 30 mg of dried PAN_A_F-FeOOH was weighed and placed in the above mixed solution for stirring. After regular sampling of 1 mL of the reaction solution, it was immediately filtered using a 0.22 μm needle filter membrane. To end the reaction, the filtrate was immediately mixed with 1 mL of methanol for quenching, and the concentration of the remaining PPOA was tested using high-performance liquid chromatography (HPLC, 1260 Infinity II, Agilent, CA, USA). Meanwhile, various parameters, including initial pH (3–9) and coexisting anions (Cl^−^, SO_4_^2−^, HCO_3_^−^, and CO_3_^2−^), influencing PPOA degradation were discussed. In addition, the stability and usefulness of PAN_A_F-FeOOH were evaluated through five cycles.

### 2.4. Phosphate Adsorption on PAN_A_F-FeOOH

Phosphate adsorption experiments used 20 mL glass bottles. The pH of KH_2_PO_4_ (20 mg P L^−1^) solution was tuned to the target value with 0.1 mol L^−1^ H_2_SO_4_ and 0.1 mol L^−1^ NaOH. Then, 10 mg of dried PAN_A_F-FeOOH was weighed and stirred in the above solution (10 mL) until the adsorption equilibrium was obtained, and the remnant phosphate concentration was tested using a visible spectrophotometer (722G, INESA, Shanghai, China). Various influencing parameters, including initial pH value (3–9), different times (0–120 min), different temperatures, different phosphate concentrations (0–50 mg P L^−1^) and coexisting anions, were also studied. Finally, the stability and practical applicability of PAN_A_F-FeOOH for phosphate removal were evaluated by continuous flow experiments and cyclic experiments.

### 2.5. Analysis and Calculation Methods

The PPOA concentration was determined by HPLC and the phosphate concentration was determined by molybdenum blue spectrophotometry. Detailed procedures and HPLC conditions are described in Appendix A. The % degradation, D (%), of PPOA by PAN_A_F-FeOOH was calculated according to Equation (1):(1)$$D=\frac{{(C}_{1}-C_{2})}{C_{1}}\times 100\%$$
where *C*_1_ and *C*_2_ are the initial concentration of PPOA and the residual concentration of PPOA after degradation, respectively.

The equilibrium adsorption capacity (*q_e_*, mg P g^−1^), adsorption capacity within a certain time (*q_t_*, mg P g^−1^), and removal efficiency *R* (%) of PAN_A_F-FeOOH for phosphate were calculated using the following equations:(2)$$q_{e}=\frac{\left(C_{0}-C_{e}\right)\times V}{m}$$
(3)$$q_{t}=\frac{\left(C_{0}-C_{t}\right)\times V}{m}$$
(4)$$R=\frac{{(C}_{0}-C_{e})}{C_{0}}\times 100\%$$
where *C*_0_, *C_e_*, and *C_t_* (mg P L^−1^) represent the initial concentration of phosphate, the equilibrium concentration of phosphate, and the phosphate concentration at a certain time, respectively. *V* (mL) is the solution volume and *m* (g) is the fiber mass.

## 3. Results and Discussion

### 3.1. Synthesis of PAN_A_F-FeOOH

As revealed in Figure 1, the synthesis of aminated polyacrylonitrile fiber (PAN_A_F) was based on our previous research [35]. Amines have strong hydrophilicity and are good metal ligands, so the grafting of PEI on PANF is conducive to the subsequent modification of iron oxides. The modification extent of PAN_A_F was calculated by the weight gain. PAN_A_F modified to different extents was prepared by regulating the substrate concentration. Then, PAN_A_F-FeOOH was constructed using the sol-gel method [38]. Compared with PANF, the quality of the aminated fiber was significantly increased (3.58–38%), indicating that the amine functional component was successfully loaded on the surface of PANF.

To investigate the effect of the amination degree of PAN_A_F on the synthesis of PAN_A_F-FeOOH, the PPOA removal ability and purification of phosphate by PAN_A_F-FeOOH synthesized from PAN_A_F with different modification degrees were tested (Appendix A). It was found that with increasing weight gain of PAN_A_F, the degradation efficiency of PAN_A_F-FeOOH for PPOA gradually increased; when the weight gain of PAN_A_F was more than 20%, the degradation efficiency was weakly increased (Appendix A). In terms of phosphate removal, with increasing PAN_A_F weight, the removal capacity for phosphate by the prepared PAN_A_F-FeOOH gradually increased, and the best weight gain of PAN_A_F was about 38% (Appendix A). The above results showed that more PEI modification was conducive to the fixation of FeOOH on the fiber support, which improved the catalytic or adsorption capacity. Therefore, the optimal weight gain of PAN_A_F was selected as 38%.

Furthermore, the PPOA degradation ability of PAN_A_F-FeOOH prepared with different concentrations of Fe(OH)_3_ colloid is demonstrated in Appendix A. At a concentration of Fe(OH)_3_ of 0.3 mol L^−1^, the degradation efficiency reached 90%, while the degradation efficiency decreased with the further increase in the concentration of Fe(OH)_3_ colloid. Appendix A also shows the phosphate adsorption capacity of PAN_A_F-FeOOH prepared with different concentrations of Fe(OH)_3_ colloid. Similar to organophosphorus degradation, the trend of phosphate removal by the fiber first increased and then decreased, with the maximum removal ability reached when the concentration of Fe(OH)_3_ colloid was 0.2 mol L^−1^. The above results indicated that the supporting of FeOOH was instrumental in improving the catalytic and adsorption capacities of the functionalized fiber. However, when the modification degree of FeOOH was too high, the application effect of PAN_A_F-FeOOH decreased. The reason was that the increased FeOOH modification on the surface of PAN_A_F within a certain range was conducive to increasing the number of effective active sites, but excessive FeOOH modification made it easier to form larger metal oxide particles [38], thus reducing the contact area between the active site and the substrate and thereby reducing its catalytic or adsorption ability. Hence, concentrations of 0.3 and 0.2 mol L^−1^ Fe(OH)_3_ colloid were selected to synthesize PAN_A_F-FeOOH for organophosphorus degradation and phosphate adsorption, respectively.

### 3.2. Characterization of PAN_A_F-FeOOH

SEM was used to characterize the surface appearance of the functionalized fiber, and the results are presented in Figure 2. Under low magnification (200 times), all of the samples were continuous and showed a complete fiber shape, indicating that the integrity of the fiber was well maintained and had not been damaged (Figure 2, trace a–f). In the SEM images under 2000 times magnification, the surface morphology of PANF was completely flat, and the diameters of PAN_A_F and PAN_A_F-FeOOH were significantly increased due to swelling in the chemical grafting reaction [35]. When the SEM images were enlarged 20,000 times, several fissures appeared on the fiber that increased the contact area between the fiber and the substrate, which was more conducive to catalysis and adsorption [33]. In addition, in the process of PAN_A_F-FeOOH synthesis, an even distribution of iron oxides could be seen on the surface of the fiber when the colloidal concentration of Fe(OH)_3_ was lower than 0.3 mol L^−^; however, when the concentration of Fe(OH)_3_ colloid was adjusted to 0.5 mol L^−1^, the iron oxide particles became larger and the uniformity was reduced.

To further demonstrate the successful synthesis of PAN_A_F-FeOOH, energy dispersive spectroscopy (EDS), Fourier-transform infrared spectroscopy (FTIR), X-ray diffraction (XRD), and thermogravimetric analysis (TG) were used, as shown in Figure 3. It was found that compared with PANF and PAN_A_F (Figure 3A, trace a,b), a new characteristic peak of the Fe element appeared in PAN_A_F-FeOOH (Figure 3A, trace c) and the peak intensity of the O element increased, indicating that iron was primarily present in the form of hydroxide on the fiber surface [29]. Changes in the functional groups on the surface of the functionalized fiber were charactered by FTIR. In the spectrum of PANF (Figure 3B, trace a), the absorption peaks at 2242 cm^−1^ and 1732 cm^−1^ could be attributed to the stretching vibration of C≡N of the first monomer acrylonitrile and C=O of the second monomer methyl acrylate [39]. Meanwhile, PAN_A_F showed a new stretching vibration peak at 1636 cm^−1^ (Figure 3B, trace b), which was caused by the formation of the amide bond and the C=O tensile vibration [40], indicating that PEI had been successfully loaded onto PANF. Furthermore, the FTIR spectrum of PAN_A_F-FeOOH (Figure 3B, trace c) presented a new stretching vibration peak 671 cm^−1^, which was caused by the stretching vibration of Fe-OH [29], indicating that FeOOH had been successfully introduced into the fiber surface. In addition, the characteristic peak of 1383 cm^−1^ belonged to NO_3_^−^ sourced from the starting Fe(NO_3_)_3_ [41], showing that some NO_3_^−^ was adsorbed on the fiber surface during the formation of FeOOH. The internal crystal structure of the fiber was analyzed by XRD (Figure 3C). The two wide diffraction peaks concentrated at 17° and 29.5° corresponded to the (100) and (110) crystal planes of the hexagonal lattice in PANF [42] (Figure 3C, trace a). PAN_A_F and PAN_A_F-FeOOH (Figure 3C, traces b,c) had similar characteristic peaks as PANF, showing that the structure of the fiber was not destroyed after functionalization. What’s more, new diffraction peaks of PAN_A_F-FeOOH were formed at 2θ = 55°, 65°, and 75°, respectively, which were similar to the characteristic peaks of FeOOH reported in the literature [43], further proving that FeOOH was truly loaded onto the fiber surface. The thermal stability of the fiber was reflected by TG (Figure 3D). PANF had no obvious quality loss up to 300 °C and still had 9.4% weight remaining at 800 °C (Figure 3D, trace a). After modification, the weight losses of PAN_A_F and PAN_A_F-FeOOH before 100 °C were mainly caused by water loss because the modified amino group had strong hydrophilicity [35]. The final residual weight of PAN_A_F-FeOOH was 41.92%, which was significantly higher than that of PANF and PAN_A_F, further illustrating the successful modification of FeOOH. The results showed that PAN_A_F-FeOOH had good thermal stability and could adapt to an aqueous environment at various temperatures.

#### Elemental Analysis (EA)

The EA results of the fiber before and after modification are displayed in Table 1. Compared with PANF, the C and N contents of PAN_A_F were significantly decreased and the H content was increased. This was because the molecular formula of PEI is (C_2_H_5_N)_n_; the proportion of C content in PEI (55.81%) was lower than that in PANF (66.17%), while the proportion of H content in PEI (11.63%) was higher than that in PANF (5.179%). The reason for the decrease in N content was that the amination process reduced the N content by converting the cyano groups in PANF to amides through hydrolysis, producing ammonia [44]. Furthermore, PAN_A_F-FeOOH had lower C, N, and H contents than PAN_A_F. The reason was that Fe and O elements were introduced into the PAN_A_F surface, which led to a decrease in the proportion of other elements.

### 3.3. Batch Experiment of PPOA Degradation by PAN_A_F-FeOOH

#### 3.3.1. Conditional Optimization of PPOA Degradation Promoted by PAN_A_F-FeOOH

The effects of pH, PAN_A_F-FeOOH dosage, PDS concentration, and reaction time on the removal of PPOA were studied (Figure 4). It was found that the pH increased from 3 to 5 and the % removal reached 90.2%. With further increase in pH (>5), the % degradation of PPOA significantly decreased to 40.9% at pH of 9 (Figure 4A). This was because the negative charge on the surface of PAN_A_F-FeOOH gradually increased with increasing pH, which increased repulsion with the PDS/PPOA anion, affecting the production of SO_4_•^−^ radicals [17]. Furthermore, PAN_A_F-FeOOH dosage also improved the removal of PPOA. The % removal without the addition of PAN_A_F-FeOOH was only 40.2%, while PPOA was nearly completely removed by increasing the mass of PAN_A_F-FeOOH to 30 mg (Figure 4B). The concentration of PDS in the system showed that increasing the concentration of PDS had a positive effect on the removal of PPOA. This was because the increased PDS concentration increased the SO_4_•^−^ concentration, which strengthened the degradation effect, and the removal effect was optimal at a concentration of 0.5 mM. However, when the PDS concentration exceeded 0.5 mM, the % removal of PPOA decreased (Figure 4C), probably due to the quenching of SO_4_•^−^ with the high concentration of PDS, which caused a decrease in the % degradation of PPOA [45], Compared with cobalt–yttrium binary oxide [11], PAN_A_F-FeOOH degraded PPOA with lower persulfate concentration. The effect of reaction time on the removal of PPOA at various temperatures was also explored (Figure 4D). The removal efficiency of PPOA gradually increased within 120 min, which could reach more than 90%. In addition, increasing temperature was conducive to the degradation reaction. This was because increasing temperature was beneficial to accelerating the reaction rate of catalytic PDS, thus increasing the generation of SO_4_•^−^ and improving the degradation efficiency of PPOA [46]. It is worth mentioning that the degradation of PPOA could be efficiently carried out at room temperature, showing high practical application value.

#### 3.3.2. Influence of Coexisting Ions and Reusability of PAN_A_F-FeOOH

The presence of coexisting anions in the aqueous environment becomes one of the important interfering factors in the activation of PDS [47]. As shown in Figure 5A, the effect of different coexisting ions on the degradation ability was investigated. While the introduction of Cl^−^ resulted in only 33.62% degradation of PPOA, this phenomenon was chiefly due to the simultaneous production of less active Cl•, Cl_2_•^−^, and HOCl products initiated by SO_4_•^−^ in the reaction process [48], which reduced the % degradation of PPOA. Positively, the coexisting anions of CO_3_^2−^, SO_4_^2−^, and HCO_3_^−^ did not significantly affect the degradation process, Compared with Co_3_O_4_-La_2_O_2_CO_3_/C-derived MOF materials [17], PAN_A_F-FeOOH was better adapted to complex ionic environments, indicating that PAN_A_F-FeOOH had potential application value in actual water bodies. In addition, the recyclability of PAN_A_F-FeOOH was explored (Figure 5B). The % removal of PPOA could reach 99% in the first three cycles, while the % removal in the fifth cycle was 78.23%, which may have been caused by the partial deactivation of the loaded catalyst and the loss of FeOOH during the degradation process [23]. The good reusability showed that PAN_A_F-FeOOH had good stability and practicability.

### 3.4. Batch Experiment of Phosphate Adsorption by PAN_A_F-FeOOH

#### 3.4.1. Influence of pH

In actual wastewater, due to its complex composition, the applicable pH range of the adsorbent is an important aspect. Therefore, the adsorption capacity of PAN_A_F-FeOOH for phosphate was measured under initial pH 3–9. As shown in Figure 6A, it was found that the phosphate removal by PAN_A_F-FeOOH increased rapidly with increasing pH value from 3 to 7, and reached the maximum value when pH = 7. Under alkaline conditions (pH 8–9), the adsorption capacity for phosphate by PAN_A_F-FeOOH slightly decreased. These results were similar to those of other Fe-based materials previously reported in the literature [49]. Additionally, this phenomenon may have been related to the dissociation equilibrium of phosphate. According to the relationship between the distribution of phosphate species and pH value, when pH = 2–3, phosphates mostly exist in the form of H_3_PO_4_, which could not be effectively adsorbed on the surface of PAN_A_F-FeOOH through electrostatic interaction and would result in a relatively low phosphate adsorption capacity. When the pH is 3–7, H_2_PO_4_^−^ and HPO_4_^2−^ are the main forms of phosphate, and these forms of phosphate were easily adsorbed by the iron site leached from the PAN_A_F-FeOOH surface. There were two main reasons for the decreased phosphate adsorption at pH 8–9. First, under alkaline conditions, PAN_A_F-FeOOH had too much negative charge on the surface, which led to the enhancement of electrostatic repulsion. Secondly, P and hydroxide ion (OH^−^) competed for adsorption sites on the PAN_A_F-FeOOH surface, resulting in a decrease in phosphate adsorption capacity [50].

#### 3.4.2. Influence of Coexisting Ions

Due to the existence of various anions (Cl^−^, NO_3_^−^, CO_3_^2−^, and SO_4_^2−^) in wastewater, the removal ability of PAN_A_F-FeOOH for phosphate may be affected. In order to verify the selectivity of PAN_A_F-FeOOH for phosphate adsorption, the purification of phosphate by PAN_A_F-FeOOH in the presence of coexisting ions was determined. As depicted in Figure 6B, the existence of Cl^−^, NO_3_^−^, and CO_3_^2−^ had little influence on phosphate adsorption by the fiber, while the presence of SO_4_^2−^ could significantly inhibit phosphate adsorption. The reason may have been the strong affinity between SO_4_^2−^ and Fe on the surface of PAN_A_F-FeOOH, which would compete with phosphate for its adsorption site, thus inhibiting phosphate adsorption [50]. Despite these findings, PAN_A_F-FeOOH still had a certain adsorption capacity for phosphate in the presence of SO_4_^2−^. This indicated that PAN_A_F-FeOOH could efficiently remove phosphate in actual water with multiple anions, showing strong practical application capability.

#### 3.4.3. Adsorption Kinetics and Thermodynamics

The study of adsorption kinetics is a key method used to study the reaction path of adsorbents. As revealed in Figure 6C, the time-dependent adsorption capacity of PAN_A_F-FeOOH for phosphate at 288, 298, and 308 K was studied. The trend of the three curves was roughly the same, with fast adsorption occurring in the first 20 min, followed by a slow adsorption rate, and reaching adsorption equilibrium after 30 min. The result demonstrated that at the beginning, the bulk of the active sites exposed on the fiber surface could quickly capture phosphate in water. With the passage of time, the adsorption process slowed down after the surface adsorption active sites of the adsorbent gradually reached saturation.

In order to better explore the adsorption mechanism of phosphate on PAN_A_F-FeOOH, the data were fitted by pseudo first-order kinetics (5) and pseudo second-order kinetics (6), respectively. The formulas of the different dynamics are as follows:(5)$$q_{e}=\left(1-e^{-K_{1}t}\right)$$
(6)$$q_{e}=\left(\frac{K_{2}q_{e}^{2}t}{1+K_{2}q_{e}t}\right)$$
where *K*_1_ and *K*_2_ are the pseudo first-order and pseudo second-order kinetics adsorption rate constants, respectively. The fitted figure and kinetic parameters are presented in Figure 6C and Appendix A, respectively. The R^2^ values of the second-order kinetic model were all higher than those of the first-order kinetic model (Appendix A), and the *q_e_* fitted by the pseudo second-order kinetics model was closer to the actual *q_e_*. The results illustrated that the adsorption process of phosphate by PAN_A_F-FeOOH was more inclined to the pseudo second-order kinetics model, showing a chemical adsorption process.

High temperatures promote inter-molecular movement and thus have a certain influence on the adsorption process, as shown in Figure 6C. It was found that the removal ability of PAN_A_F-FeOOH for phosphate increased with rising temperature, which showed that the adsorption process was an endothermic reaction. Furthermore, the thermodynamic parameters of Δ*S*^*o*^ (standard entropy, J mol^−1^ K^−1^), Δ*H*^*o*^ (standard enthalpy, kJ mol^−1^), and Δ*G*^*o*^ (Gibbs free energy, kJ mol^−1^) were further used to reveal the relationship between temperature and phosphate adsorption, which were calculated using the following equations:(7)$$K_{c}=\frac{q_{e}}{C_{e}}$$
(8)$$∆G^{o}=-RTlnK_{c}$$
(9)$$lnK_{c}=\frac{∆S^{o}}{R}-\frac{∆H^{o}}{RT}$$
where *K_c_* was the distribution coefficient, *R* was the gas constant (8.314 J mol^−1^ k^−1^), and *T* was the temperature (K), respectively. The Δ*S*^*o*^ and Δ*H*^*o*^ could be calculated by plotting *lnK_c_* versus 1/*T*, and the relevant curves and calculated parameters are displayed in Appendix A, respectively. The Δ*G*^*o*^ was negative, showing that the removal of phosphate by PAN_A_F-FeOOH adsorbent was spontaneous. Furthermore, the absolute value of Δ*G*^*o*^ increased with rising temperature, which meant that the higher the temperature, the more natural the reaction. Δ*H*^*o*^ values were positive, suggesting that phosphate adsorption by PAN_A_F-FeOOH is an endothermic reaction. In addition, a positive Δ*S*^*o*^ value meant that the randomness of the solid/solution interface increased irregularly during phosphate adsorption.

#### 3.4.4. Adsorption Isotherm

In order to reveal the adsorption capacity of PAN_A_F-FeOOH for phosphate, isothermal adsorption experiments were carried out at room temperature with different initial phosphate concentrations. As shown in Figure 6D, the adsorption ability of PAN_A_F-FeOOH for phosphate increased with the initial concentration of phosphate until the adsorption saturation was reached. This may have been due to the presence of more coordination sites for phosphorus at lower concentrations, and the active adsorption sites became saturated with increasing phosphorus concentrations. Moreover, Langmuir (10) and Freundlich (11) adsorption isotherm models were used to fit the adsorption data, the equations of which are as follows:(10)$$q_{e}=\frac{K_{l}q_{m}C_{e}}{1+K_{L}C_{e}}$$
(11)$$q_{e}=K_{F}C_{e}^{1/n}$$
where *q_m_* is the theoretical maximum adsorption estimated by the Langmuir model, *K_L_* is the Langmuir constant, and *K_F_* and n are the Freundlich constant and homogeneity coefficient, respectively. The graphs and related data obtained by fitting curves are located in Figure 6D and Appendix A, respectively.

The result showed that the R^2^ values of the fitting curve of the Langmuir model were larger than those of the Freundlich model (Appendix A), which indicated that the Langmuir model was more suitable for the adsorption of phosphate by PAN_A_F-FeOOH. Therefore, the adsorption of phosphate by the fiber tended to be uniform monolayer adsorption. It is worth mentioning that the maximal adsorption amount calculated by Langmuir was 9.92 mg P g^−1^, which was higher than that reported in other literature for iron-modified composite phosphorus adsorbents such as solid carbon source/zero-valent iron composite (0.21 mg P g^−1^) [51], Fe-modified corn straw biochar (0.56 mg P g^−1^) [52], and iron-containing activated carbon (8.47 mg P g^−1^). Hence, PAN_A_F-FeOOH showed great potential for phosphate removal from water.

#### 3.4.5. Reusability and Continuous Flow Application of PAN_A_F-FeOOH

To explore the regeneration capacity of PAN_A_F-FeOOH as a phosphate removal adsorbent, PAN_A_F-FeOOH saturated with phosphate was eluted with 0.05 mol L^−1^ NaOH for 1 h. As shown in Figure 6E, the % phosphate removal by PAN_A_F-FeOOH tenuously decreased after 5 cycles, and the fiber still maintained a % phosphate removal of more than 88%. This result demonstrated that PAN_A_F-FeOOH had excellent stability and reusability.

Furthermore, a continuous flow experiment was carried out to verify the practical application capability of PAN_A_F-FeOOH. The continuous flow device is shown in Appendix A. Dried PAN_A_F-FeOOH (300 mg) was placed in a silicone tube (length 100 mm, diameter 5.6 mm), Chaohu Lake water with an initial concentration of 1 mg P L^−1^ was injected at a flow rate of 1 mL min^−1^ using a peristaltic pump (the concentration of Chaohu Lake water was specially adjusted to 1 mg P L^−1^), and then the concentration of phosphate in the receiving solution was determined. The penetration curves of PAN_A_F-FeOOH for phosphate removal were plotted (Figure 6F), and it was found that with increasing phosphate outflow liquid volume, its concentration gradually increased. When the volume of liquid was less than 280 mL, the % phosphate removal could be maintained above 95%. This result suggested that PAN_A_F-FeOOH could effectively purify phosphate under continuous flow conditions. Repeating the fiber multiple times could be used as an important building material to enhance the mechanical strength of cement [53].

### 3.5. Mechanisms of PPOA Degradation and Phosphate Adsorption by PAN_A_F-FeOOH

In this work, reactive oxygen species (ROS) were determined by electron paramagnetic resonance (EPR) to describe the key free radicals in the PPOA degradation process (Appendix A). The EPR results showed that the signal strength of SO_4_•^−^ and •OH catalyzed by PAN_A_F-FeOOH was significantly improved when compared with the pure PDS system. Additionally, the peak intensity of SO_4_•^−^ was higher than that of •OH (Appendix A), while the strength of singlet oxygen ^1^O_2_ had no obvious change (Appendix A). The above phenomena explained that SO_4_•^−^ and •OH were the two dominant active species for PPOA degradation. The reason was that SO_4_•^−^ could be slowly generated by dissociation of the peroxide bond [54], and Fe^3+^ on the fiber catalyst could react with S_2_O_8_^2−^ to produce Fe^2+^ and the persulfate radical (S_2_O_8_^−^•) [25]. The newly produced Fe^2+^ could activate persulfate to produce the sulfate radical [55], hence Fe^2+^ was continuously produced through the above electron transfer reaction, which improved the efficiency of SO_4_•^−^ generation. In the PPOA degradation system, SO_4_•^−^ was preferentially produced, then it reacted with H_2_O to produce •OH. In addition to ROS participating in the degradation of PPOA, the enrichment ability of PAN_A_F-FeOOH for adsorption of PPOA was also an important factor for efficient catalysis. Relevant research shows that improving the affinity and adsorption capacity of the catalyst for the substrate is conducive to improving its catalytic capacity [56,57]. To verify this assumption, the adsorption capacity of PAN_A_F-FeOOH for PPOA was studied (Appendix A). PANF had little enrichment capacity for PPOA adsorption; however, after amination and FeOOH modification, the % removal of PPOA by the functionalized fiber was obviously increased to 40% and 45%, respectively. Similar to our previous research [31], the above results showed that the functionalized modification of the fiber support created a microenvironment on the surface of the fiber that was conducive to the entry of PPOA, which made it easier to degrade by the ROS generated by the activated persulfate of PAN_A_F-FeOOH. This was because the existence of the microenvironment effect shortened the diffusion distance of reactive oxygen radicals, thus accelerating the degradation of PPOA.

Under optimal degradation conditions, when the degradation reaction was over, the inorganic phosphorus and total phosphorus in the solution were determined by molybdenum blue colorimetry and ICP-OES (ICP-MS, Thermo scientific, Waltham, MA, USA), respectively. The results showed that there was a small amount of phosphate in the solution and the concentration of total phosphorus was about 10% of the original concentration, indicating that PAN_A_F-FeOOH successfully degraded PPOA and simultaneously absorbed the degradation product of inorganic phosphorus. Hence, the phosphate adsorption mechanism of PAN_A_F-FeOOH was further studied by X-ray photoelectron spectroscopy (XPS), as shown in Figure 7, and the atomic percentage of the fiber is presented in Appendix A. As depicted in Figure 7A, similar to the results of EDS, a new peak of Fe emerged in the PAN_A_F-FeOOH spectrum, indicating the successful grafting of FeOOH on the fiber (Figure 7A, trace c). After adsorption of phosphate, the high-resolution spectrum of PAN_A_F-FeOOH-P possessed the characteristic peak of P 2p (Figure 7B), which was direct evidence of phosphate adsorption by PAN_A_F-FeOOH.

The high-resolution spectrum of Fe 2p for PAN_A_F-FeOOH was parsed into 2p_1/2_ at 724.4 eV and 2p_3/2_ at 712.4 eV (Figure 7C, trace a). The binding energies of Fe 2p for PAN_A_F-FeOOH-P (Figure 7C, trace b) were weakly reduced to 724.3 and 712.1 eV, which was due to the formation of the Fe–O–P bond between phosphate and Fe modified on the fiber, resulting in the increased electron cloud density of the atom [58]. The results indicated that the iron modified on the fiber surface played an significant role in phosphate adsorption. Furthermore, the high-resolution spectrum of N 1s was decomposed into 401.6, 400.0, and 399.2 eV, which were the characteristic peaks of O≡C–N, –NH–/–NH_2_, and –NR_2_, respectively [33]. After adsorption of phosphate, the binding energies of the N1s spectrum slightly increased because the protonated amine group on the fiber surface combined with phosphate through electrostatic force [59]. Hence, the removal mechanism of phosphate by PAN_A_F-FeOOH-P was mainly due to the strong binding force of iron and the electrostatic force of protonated amine.

### 3.6. Comparison of Different FeOOH Phosphate Adsorbents

The comparison of phosphate adsorption by the fiber was performed as follows. In this study, the adsorption capacity, cycling performance, and equilibrium time of PAN_A_F-FeOOH were compared with other adsorbents for phosphate removal (Appendix A). PAN_A_F-FeOOH had a better phosphate adsorption capacity than other adsorbents (Appendix A, entries 1 and 5), and the cycling performance of the fiber (5 times) was higher than that of other adsorbents. Furthermore, PAN_A_F-FeOOH reached adsorption equilibrium (30 min) faster than the other adsorbent materials. Overall, PAN_A_F-FeOOH showed high phosphate removal efficiency, good adsorption capacity, and excellent reusability.

### 3.7. Practical Application Capability Test of PAN_A_F-FeOOH

This study also tested the practical application capability of PAN_A_F-FeOOH to remove phosphorus in actual water and simulated wastewater, respectively. The total phosphorus concentration was adjusted to about 1 mg P L^−1^ (0.5 mg P L^−1^ PPOA + 0.5 mg P L^−1^ KH_2_PO_4_) in both real water (Chaohu Lake, China) and deionized water. Dried 50 mg of PAN_A_F-FeOOH and 0.5 mmol L^−1^ of PDS were added to 20 mL of the above water samples, and the remaining total phosphorus concentration was measured after stirring for 1 h. The total phosphorus concentration in the water samples before and after fiber treatment were measured. The results showed that PAN_A_F-FeOOH could effectively remove 83.6% of total phosphorus in the water sample of Chaohu Lake, while the % removal reached 90.4% in the simulated wastewater. The application capability of the fiber in real water bodies was slightly lower than that in simulated wastewater samples, possibly due to complex coexisting ion interference in real water bodies. Altogether, PAN_A_F-FeOOH showed excellent application ability for phosphorus removal in real water.

## 4. Conclusions

In this work, a novel FeOOH-modified polyacrylonitrile fiber (PAN_A_F-FeOOH) was intelligently synthesized for the simultaneous removal of organophosphorus and phosphate. PAN_A_F with a weight gain of 38% and iron concentrations of 0.2 and 0.3 mol L^−1^ were the optimal synthesis conditions for organophosphorus degradation and phosphate adsorption, respectively, by the PAN_A_F-FeOOH. The degradation conditions for phenylphosphonic acid (PPOA) were optimized, and the PAN_A_F-FeOOH (30 mg) was able to activate PDS (0.5 mmol L^−1^) to remove more than 99% of PPOA (0.5 mg L^−1^, pH = 5) within 1.5 h. Positively, phosphate produced by the degradation process could be efficiently adsorbed by the PAN_A_F-FeOOH. Hence, the phosphate adsorption by the PAN_A_F-FeOOH was investigated. The fiber showed wide pH adaptability with a maximum removal capacity of 9.92 mg P g^−1^, which was higher than most reported iron-based functional materials. Furthermore, the PAN_A_F-FeOOH possesses the advantages of anti-interference with coexisting ions, excellent recyclability (5 times), and continuous flow application. In addition, electron paramagnetic resonance and X-ray photoelectron spectroscopy were used to clarify the promotion mechanism of the fiber surface microenvironment. It was found that SO_4_•^−^ and •OH were the key radicals for PPOA degradation, and the strong enrichment ability of the functional moieties on the fiber surface also accelerated the removal of PPOA. In addition, the phosphate removal mechanism was chiefly attributed to the synergy of Fe and amine groups. This research provides new ideas for the simultaneous removal of organic and inorganic phosphorus, which is of great significance for pollution remediation and the reuse of organic phosphorus.

## Figures and Tables

**Figure 1 polymers-15-01918-f001:**
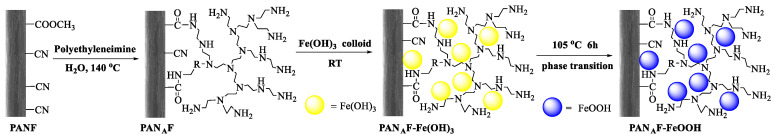
The preparation of FeOOH-loaded polyacrylonitrile fiber.

**Figure 2 polymers-15-01918-f002:**
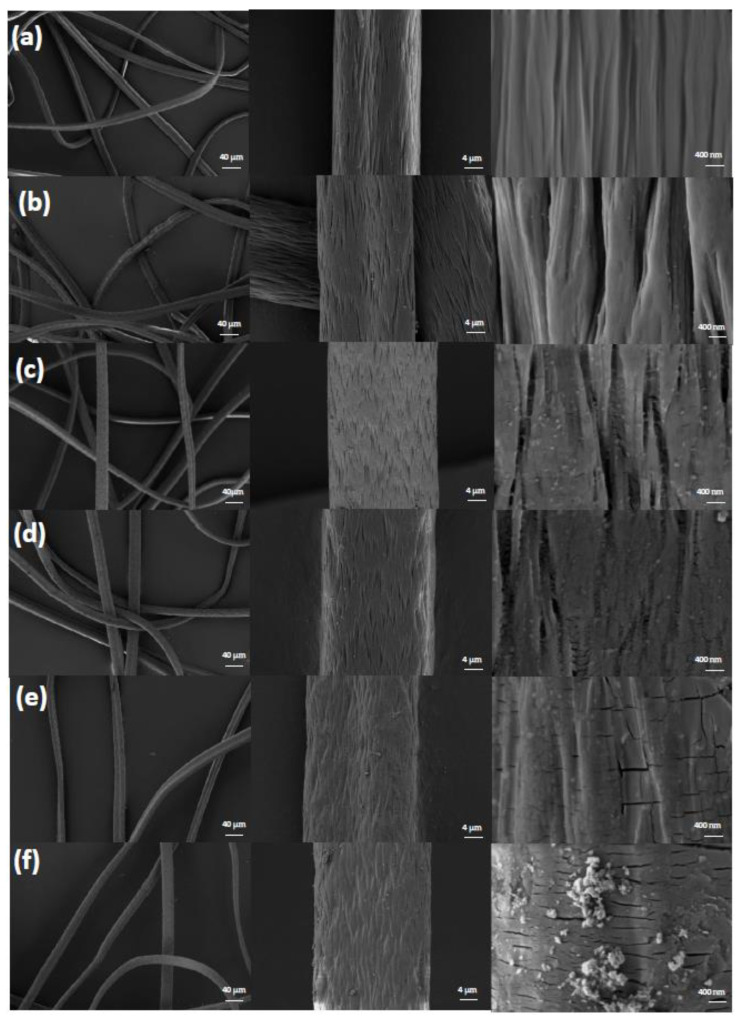
The SEM photographs of (**a**) PANF, (**b**) PAN_A_F, and (**c**–**f**) PAN_A_F-FeOOH prepared with Fe(OH)_3_ colloid at 0.05, 0.2, 0.3, and 0.5 mol L^−1^, respectively. Magnifications are 200, 2000, and 20,000 times.

**Figure 3 polymers-15-01918-f003:**
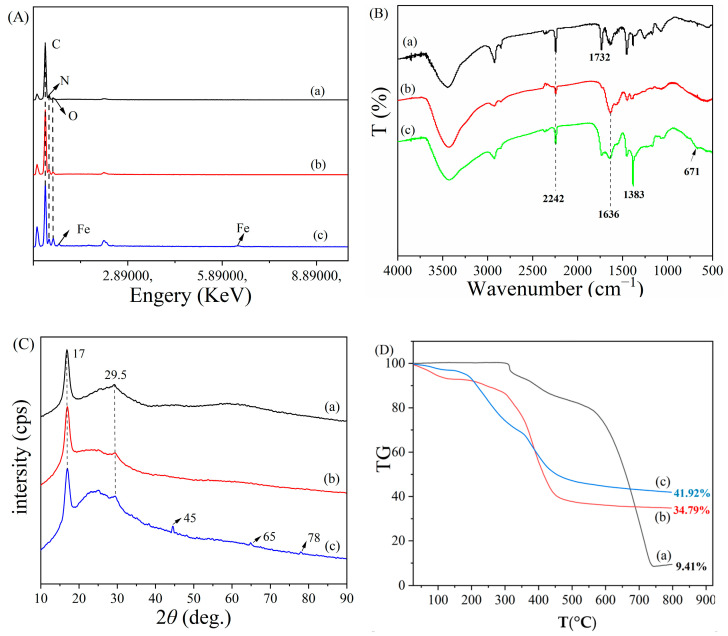
(**A**) The EDS spectra of (a) PANF, (b) PAN_A_F, and (c) PAN_A_F-FeOOH; (**B**) The FTIR spectra of (a) PANF, (b) PAN_A_F, and (c) PAN_A_F-FeOOH; (**C**) The XRD patterns and (**D**) TGA spectra of (a) PANF, (b) PAN_A_F, and (c) PAN_A_F-FeOOH.

**Figure 4 polymers-15-01918-f004:**
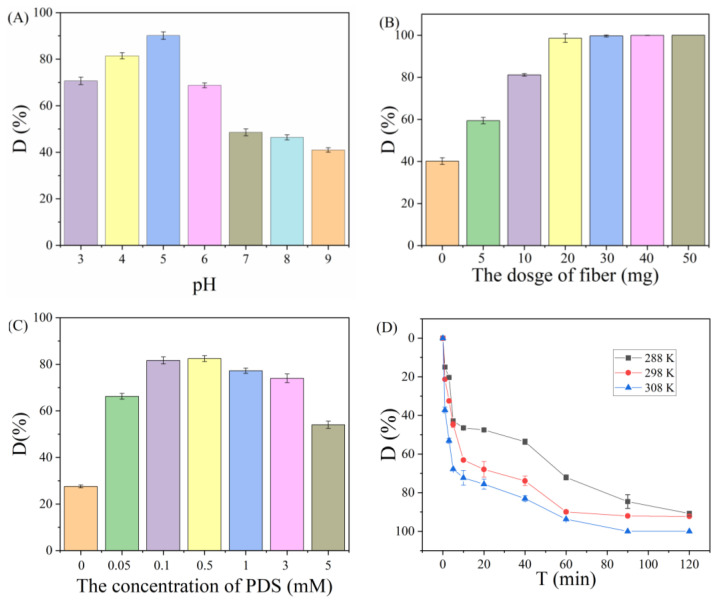
Influence of (**A**) pH, (**B**) fiber dosage, (**C**) PDS concentration, and (**D**) reaction time on PPOA degradation.

**Figure 5 polymers-15-01918-f005:**
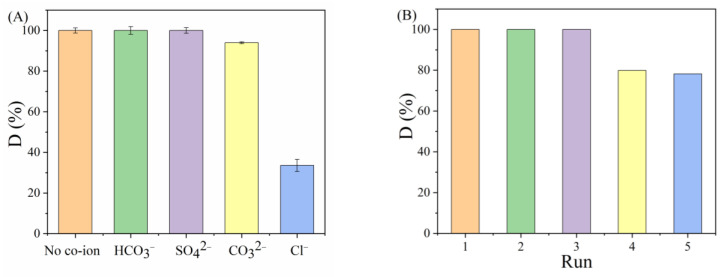
(**A**) The impact of coexisting ions on PPOA degradation; (**B**) renewability of PAN_A_F-FeOOH for PPOA degradation.

**Figure 6 polymers-15-01918-f006:**
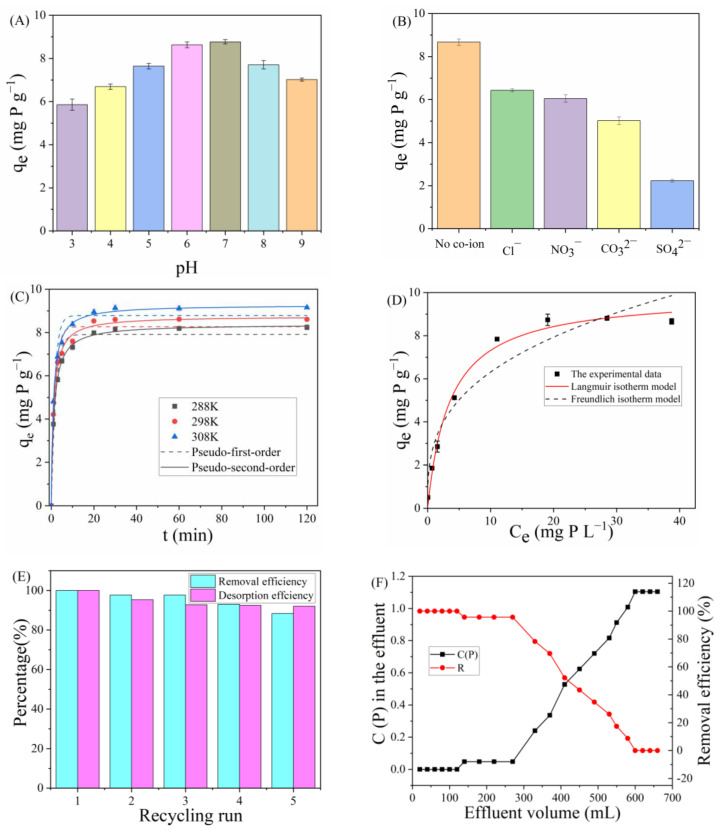
The influence of (**A**) pH, (**B**) coexisting anions, (**C**) contact time, and (**D**) initial phosphate concentration on phosphate removal by PAN_A_F-FeOOH. (**E**) Reusability of PAN_A_F-FeOOH for phosphate removal and (**F**) breakthrough curves of phosphate solutions in continuous flow conditions.

**Figure 7 polymers-15-01918-f007:**
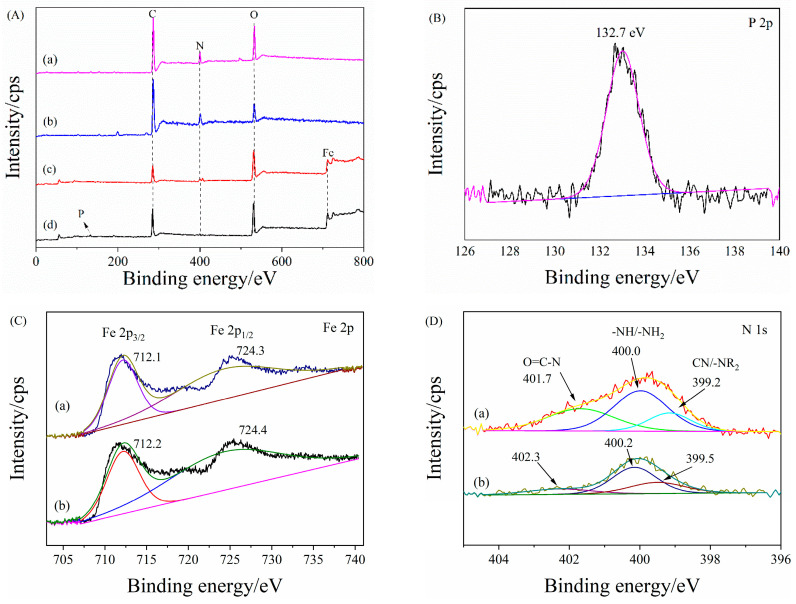
(**A**) XPS survey of (a) PANF, (b) PAN_A_F, (c) PAN_A_F-FeOOH, and (d) PAN_A_F-FeOOH-P; (**B**) High-resolution spectrum of P 2p for PAN_A_F-FeOOH-P; (**C**) High-resolution Fe 2p spectrum of (a) PAN_A_F-FeOOH and (b) PAN_A_F-FeOOH-P; (**D**) High-resolution N 1s spectrum of (a) PAN_A_F-FeOOH and (b) PAN_A_F-FeOOH-P.

**Table 1 polymers-15-01918-t001:** Elemental analysis data for PANF, PAN_A_F, and PAN_A_F-FeOOH.

Entry	Sample	C (%)	N (%)	H (%)
1	PANF	66.37	24.86	5.18
2	PAN_A_F	55.14	20.15	6.57
3	PAN_A_F-FeOOH	47.32	18.33	4.59

## Data Availability

The data presented in this work are available upon request from the corresponding author.

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
