# Peer review of "Synthesis of FeOOH-Loaded Aminated Polyacrylonitrile Fiber for Simultaneous Removal of Phenylphosphonic Acid and Phosphate from Aqueous Solution"

_polymers, 2023, doi:10.3390/polym15081918_

Round 1

Reviewer 1 Report

I carefully reviewed this manuscript. This manuscript has many interesting results and aspects. I think It will be published, if the followings are revised and modified.

I have some comments on the contents of this manuscript

Line 23: “could effectively activate” is change to “effectively activated”.

Line 28: According to Figure 1, the adsorbents in this study is prepared from PAN chains. Therefore, it is difficult to form a monolayer. It should be revised as follows. For example, the expression of Langmuir isotherm shows the 1:1 stoichiometric interaction of A and B.

Line 42: A space is inserted between the word and reference number. Same for the others. “conversion[1] is changed to “conversion [1]”.

Line 107: “Polyethylene imine” is changed to “Polyethylene imine”.

Line 117: For “Figure 1.:”, “:” is removed.

Line 120: For the representation of temperature, the space between the figure and unit is not required. “60°C” is changed to “60°C”. Same for the others.

Line 128: Show the temperatures before and after cooling.

Line 146: “adsorption by PANAF-FeOOH” is changed to “adsorption on PANAF-FeOOH”

Line 148: “L-1” is changed to superscript.

Line 167: Is Eq.(2) required?

Line 179: “with different modification extent” is changed to  “with different modification extents”

Line 183: What does the term “appropriate degree” mean? I think it is revised or explained more.

Line 195: It should be revised to “at the concentration of Fe(OH)3 of 0.3 mol L-1”.

Section 3.1: I think the reaction scheme of degradation is required.

Table 1: More explanation is required on the title. It is too simple. The space between lines is removed.

Line 283: “the degradation rate” should be changed to “the degradation %” because the term “rate” has the meaning of velocity.

Figure 4: The scale of y axis should be revised. Start from 0 (zero).

It is important. From Figure 4(D), for adsorption at 288K”, the equilibrium is not reached. The constant value is not obtained.

Figure 4(c): On the involvement of PDS to the PPOD degradation %, more explanations are required. Why did the degradation % increased and decreased at higher PDS concentrations?

Figure 6: The scale of y axis should be unified, such as from 0 (zero) to 10 (mg P/g). There are two units, (mg P/g) and (mg P g-1). It should be unified. The scale of x axis should start from 0 (zero) in (C) and (D).

Line 493-506: The binding energy is too fine. For example, 724.372 eV should be revised to 724.4 eV. Same for the others.

Removal of PPOA and phosphate is not compared to others.

Reference: The representation of author name is mixed. The manes of all authors should be described according to the instructions. In addition, the name of each journal should be properly abbriviated.

Author Response

The response to the reviewer 1: (In blue in the revised manuscript)

Comments to Author: I carefully reviewed this manuscript. This manuscript has many interesting results and aspects. I think it will be published, if the followings are revised and modified. I have some comments on the contents of this manuscript

Question 1: Line 23: “could effectively activate” is change to “effectively activated”

Response: Thanks for your modification. The “could effectively activate” has been revised as “effectively activated”.

Question 2: Line 28: According to Figure 1, the adsorbents in this study is prepared from PAN chains. Therefore, it is difficult to form a monolayer. It should be revised as follows. For example, the expression of Langmuir isotherm shows the 1:1 stoichiometric interaction of A and B.

Response: Thank you for your comments. After consulting relevant materials, the functional molecules on the surface of a solid have a residual valence force outward, and when that residual valence force acts in a range comparable to the molecular diameter, adsorption of a single molecular layer occurs on the surface of the adsorbent. In this work, although PANF has thousands of molecular chains, the modification of fibers mainly occurs on the surface layer. Therefore, the removal of phosphate through chemical adsorption mainly occurs on the surface layer of the fiber. Through isothermal adsorption model fitting, the adsorption data is more inclined towards the Langmuir adsorption model, which is closer to single-layer adsorption[1]. Furthermore, not all modified functional groups are involved in the removal of phosphate, we think that it is reasonable to summarize the adsorption process as monolayer adsorption.

Question 3: Line 42: A space is inserted between the word and reference number. Same for the others. “conversion[1] is changed to “conversion [1]”.

Response: We have carefully reviewed and revised the problem you mentioned in the paper.

Question 4: Line 107: “Polyethyleneimine” is changed to “Polyethylene imine”.

Response: This was an oversight on our part and it has now been corrected

Question 5: Line 117: For “Figure 1.:”, “:” is removed.

Response: The problem has been corrected.

Question 6: Line 120: For the representation of temperature, the space between the figure and unit is not required. “60␣°C” is changed to “60°C”. Same for the others.

Response: We have revised the manuscript by removing all spaces.

Question 7: Line 128: Show the temperatures before and after cooling.

Response: Thank you for your suggestion, we have added the temperature in the manuscript.

Question 8: Line 146: “adsorption by PANAF-FeOOH” is changed to “adsorption on PANAF-FeOOH”

Response: “adsorption by PANAF-FeOOH” has been revised as “adsorption on PANAF-FeOOH”.

Question 9: Line 148: “L-1” is changed to superscript.

Response: Thanks for your suggestion. We have revised the manuscript

Question 10: Line 167: Is Eq.(2) required?

Response: As you mentioned, removing Eq.(2) does not have much impact. The concepts of qe and qt are different. The former refers to the equilibrium adsorption capacity, while the latter refers to the adsorption capacity at a certain time t. When t is equal to the equilibrium time, the qe and qe are equal. In order to provide readers with more intuitive guidance on the difference between qe and qt, we think Eq.(2) is still necessary in the paper.

Question 11: Line 179: “with different modification extent” is changed to “with different modification extents”.

Response: We have revised the manuscript to change “with different modification extent” to “with different modification extents”.

Question 12: Line 183: What does the term “appropriate degree” mean? I think it is revised or explained more.

Response: Thanks for your comment. The modification degree of PANAF can be prepared by adjusting the concentration of PEI, within a certain PEI concentration range, the modification degree of PANAF increases with increasing PEI concentration. Hence, the “appropriate degree” in this work means “appropriate modification degree of amination”. For better understanding by readers, we have made the following modifications: To investigate the effect of amination degree of PANAF on the synthesis of PANAF-FeOOH, the removal ability of PPOA and purification of phosphate by PANAF-FeOOH synthesized from PANAF with different modification degree were tested (Figure. S1).

Question 13: Line 195: It should be revised to “at the concentration of Fe(OH)3 of 0.3 mol L-1”.

Response: We have made modifications according to your suggestion.

Question 14: Section 3.1: I think the reaction scheme of degradation is required.

Response: Thank you for your suggestion. The FeOOH on the fiber surface reacts with PDS to form SO4- radicals, and SO4- reacts with H2O to form ·OH radicals. The seactive oxygen species (ROS) attacks PPOA for degradation reaction, the products are phosphate, CO2 and H2O. The reaction process is relatively simple, so we did not include it in the main text.

Question 15: Table 1: More explanation is required on the title. It is too simple. The space between lines is removed.

Response: We explained the table name in detail and changed it to "Elemental analysis data for PANF, PANAF and PANAF-FeOOH. ". We have also modified the table to remove the redundant parts, thank you for your suggestion.

Question 16: Line 283: “the degradation rate” should be changed to “the degradation %” because the term “rate” has the meaning of velocity.

Response: We agree with your opinion. The "the degradation rate" has been revised as "the degradation %" in the entire manuscript.

Question 17: Figure 4: The scale of y axis should be revised. Start from 0 (zero)

Response: We have revised the Figure 4A in the manuscript based on your suggestion.

Question 18: It is important. From Figure 4(D), for adsorption at 288K”, the equilibrium is not reached. The constant value is not obtained.

Response: Thank you for your comment. We have verified the data and it is indeed the case. The curve of 288K is not balanced but close to equilibrium. Furthermore, through the degradation curves at the different temperatures, it can be clearly seen that the degradation ability increases with increasing temperature. Therefore, the experimental results can well illustrate the impact of temperature, and the curves at different temperatures can support the viewpoint in the paper.

Question 19: Figure 4(c): On the involvement of PDS to the PPOD degradation %, more explanations are required. Why did the degradation % increased and decreased at higher PDS concentrations?

Response: Increases of PDS concentration will lead to increased SO4•- concentration generated by catalyzed, which helps degrade effects. However, high-concentration PDS can cause self -quenching SO4•- free radicals, which is the main reason that affects degradation. We have added a more detailed description in the manuscript: “The concentration of PDS in the system showed that increasing the concentration of PDS had a positive effect on the removal of PPOA, this is because the increase in the PDS concentration increases the SO4- concentration, which strengthens the degradation effect, and the removal effect was optimal at a concentration of 0.5 mM. However, when the PDS concentration exceeded 0.5 mM, the removal % of PPOA decreased (Figure. 4C), probably due to the quenching of SO4- in the high concentration of PDS, which caused a decrease in the degradation rate of PPOA [48].”

Question 20: Figure 6: The scale of y axis should be unified, such as from 0 (zero) to 10 (mg P/g). There are two units, (mg P/g) and (mg P g-1). It should be unified. The scale of x axis should start from 0 (zero) in (C) and (D)

Response: We have revised the Figure 4 in the manuscript based on your suggestions.

Question 21: Line 493-506: The binding energy is too fine. For example, 724.372 eV should be revised to 724.4 eV. Same for the others.

Response: We agree with your proposal, and binding energy data in the text has been modified as required. The Figures were also revised.

Question 22: Removal of PPOA and phosphate is not compared to others.

Response: Thanks to your suggestions, we have revised the part of the manuscript that compares other materials to remove PPOA and phosphate, as follows: (1) Positively, the coexisting anions of CO32-, SO42- and HCO3- did not significantly affect the degradation process, Compared with Co3O4-La2O2CO3/C-derived MOFs materials [47], PANAF-FeOOH is better adapted to complex ionic environments, indicating that the PANAF-FeOOH has potential application value in actual water body. (2) Compared with cobalt-yttrium binary oxide [50], the PANAF FeOOH degrades PPOA with lower persulfate concentration.

The comparison of phosphate adsorption by fibers is as follows: In this study, the adsorption capacity, cycling performance, and equilibrium time of PANAF-FeOOH were compared with other adsorbents for phosphate removal (Table S5.) PANAF-FeOOH had better phosphate adsorption capacity than other adsorbents (Tables S5, entries 1 and 5), and the cycling performance of the fiber (5 times) was higher than other adsorbents. Furthermore, PANAF-FeOOH reached adsorption equilibrium time (30 min) faster than the other adsorbent materials. Overall, the PANAF-FeOOH shows a high phosphate removal efficiency, good adsorption capacity and excellent reusability.

Question 23: Reference: The representation of author name is mixed. The manes of all authors should be described according to the instructions. In addition, the name of each journal should be properly abbriviated.

Response: We have revised the References in the manuscript based on your suggestions.

Reviewer 2 Report

The article provides valuable information on the synthesis and characterization of FeOOH-loaded aminated polyacrylonitrile fiber (PANAF-FeOOH) and its application as an adsorbent for the simultaneous removal of organic (phenylphosphonic acid) and inorganic phosphorus (phosphate) from wastewater. The conducted research is of great importance to the pollution remediation and reuse of organic phosphorus.

According to the presented results and discussion, the article is well structured. However, some facts require clarification and revision:

1. Investigations of the phosphate sorption process were performed on simulated aqueous solutions of phosphorous forms, not on real wastewater. Therefore, I propose to make a change in the title: instead of „wastewater“, it should be „aqueous solution“.

2. Please verify the truth of the following statement (page 6, lines 239-241): „the FTIR spectrum of PANAF-FeOOH (Figure. 3B, trace c) presented a new stretching vibration peak at 1383 cm-1 and 671 cm-1, respectively, which may be caused by the Fe-N bond formed between iron and amine group and the stretching vibration of Fe-OH  [28]“.

On what basis do the authors state that the vibration at 1383 cm-1 originates from the F-N bond (formed between iron and amine group)? The cited reference [28] does not refer to the explanation of the Fe-N bond vibration, but only confirms the FeOOH form (Fe-OH vibration). This should be clarified and confirmed by some other reference.

Namely, the F-N vibration appears in the FTIR region of 400-500 cm-1. The IR vibration at 1385 cm−1 is typical of the NO3− ion (probably from the starting Fe(NO3)3. It can also appear in the case of stretching vibration of the C-N bond.

3. From the aspect of practical application of the sorbent in real systems, the end of the cycle of used sorbent is important. The manuscript does not say anything about the fate of the new sorbent after complete exploitation. So please add some content about it, as well as environmental impact.

Author Response

The response to the reviewer 2: (In green in the revised manuscript)

The article provides valuable information on the synthesis and characterization of FeOOH-loaded aminated polyacrylonitrile fiber (PANAF-FeOOH) and its application as an adsorbent for the simultaneous removal of organic (phenylphosphonic acid) and inorganic phosphorus (phosphate) from wastewater. The conducted research is of great importance to the pollution remediation and reuse of organic phosphorus. According to the presented results and discussion, the article is well structured. However, some facts require clarification and revision:

Question 1: Investigations of the phosphate sorption process were performed on simulated aqueous solutions of phosphorous forms, not on real wastewater. Therefore, I propose to make a change in the title: instead of wastewater, it should be “aqueous solution”.

Response: Good suggestion, we have corrected the title of the manuscript as well as the main manuscript according to your suggestion.

Question 2: Please verify the truth of the following statement (page 6, lines 239-241): “the FTIR spectrum of PANAF-FeOOH (Figure. 3B, trace c) presented a new stretching vibration peak at 1383 cm-1 and 671 cm-1, respectively, which may be caused by the Fe-N bond formed between iron and amine group and the stretching vibration of Fe-OH [28]”.

Response: Thank you for your valuable comment. We do not have direct evidence to suggest that 1383 cm-1 is a Fe-N stretching vibration, and the statement in the paper is a speculation, which is indeed not scientific enough. Based on your suggestion and literature review, 1383 cm-1 is the characteristic peak of NO3-. We have made the following modifications to the paper:Furthermore, the FTIR spectrum of PANAF-FeOOH (Figure. 3B, trace c) presented a new stretching vibration peak 671 cm-1, which is caused by the stretching vibration of Fe-OH, the result indicates that FeOOH has been successfully introduced into the fiber surface. Besides, the characteristic peak of 1383 cm-1 belongs to NO3- sourced from starting Fe(NO3)3, showing that some NO3- is adsorbed on the fiber surface during the formation of FeOOH.

Question 3: From the aspect of practical application of the sorbent in real systems, the end of the cycle of used sorbent is important. The manuscript does not say anything about the fate of the new sorbent after complete exploitation. So please add some content about it, as well as environmental impact.

Response: The adsorbent remains valuable after multiple cycles, for example as an important building material for improving the mechanical strength of cement. This section was added to the manuscript: The fiber that has been repeated multiple times could be used as an important building material to enhance the mechanical strength in the cement.

Reviewer 3 Report

The study entitled “Synthesis of FeOOH loaded aminated polyacrylonitrile fiber for simultaneous removal of phenylphosphonic acid and phosphate from wastewater” proposed is to degradation of organic phosphorus (OP) and synchronous recovery the produced inorganic phosphorus is the key to the utilization of OP resource and the prevention of water eutrophication. Furthermore, the PANAF-FeOOH possessed excellent phosphate removal capacity with maximal adsorption quantity of 9.92 mg P g-1. The adsorption kinetics and isotherms of PANAF-FeOOH on phosphate were better depicted by pseudo-quadratic kinetics and Langmuir isotherm models, showing a monolayer chemisorption procedure. Inconclusion, this study provides evidence for PANAF-FeOOH as a potential material for the degradation of OP and simultaneous recovery of phosphate. There is potential in the research presented in the article, and it should be published after the following revisions have been made:

1.      The abstract would benefit from a clearer statement of the research objectives. While the authors briefly mention that their goal is to enhance the removal of organic phosphorus and phosphate, it would be helpful to state explicitly why this is important (e.g. to prevent water eutrophication and utilize OP resources).

2.      The results section of the abstract could be restructured to make it easier to understand. Consider separating the discussion of OP degradation and phosphate removal into two separate paragraphs, and clearly stating the specific findings for each (e.g. the best FeOOH concentration for OP degradation and phosphate adsorption).

3.      The introduction provides a clear overview of the significance of phosphorus and its pollution problems. However, it would be helpful to provide some quantitative data on the extent of the problem and its impact on the environment and human health.

4.      Please provide more details on the continuous flow test, such as the flow rate and the concentration of PPOA and phosphate used in the test.

5.      Please provide more details on the specific procedure and conditions used for the measurement.

6.      Please provide more details on the characterization techniques used to analyze the synthesized materials. For instance, what was the wavelength range used for the FTIR spectra? What was the scanning range for the XRD analysis? What was the step size and scanning rate for the TGA analysis?

7.      In the Results and discussion section, it would be helpful to discuss the practical implications of the results. For example, what are the potential applications of the FeOOH loaded aminated polyacrylonitrile fiber in real-world wastewater treatment? Are there any limitations or challenges that need to be addressed before the material can be used on a larger scale?

8.      It would be helpful if the authors provided more information on the potential practical applications of their fiber material. For example, they could discuss how the material could be used in wastewater treatment plants or other industrial processes to remove organic and inorganic phosphorus from wastewater.

Author Response

The response to the reviewer 3: (In red in the revised manuscript)

The study entitled “Synthesis of FeOOH loaded aminated polyacrylonitrile fiber for simultaneous removal of phenylphosphonic acid and phosphate from wastewater” proposed is to degradation of organic phosphorus (OP) and synchronous recovery the produced inorganic phosphorus is the key to the utilization of OP resource and the prevention of water eutrophication. Furthermore, the PANAF-FeOOH possessed excellent phosphate removal capacity with maximal adsorption quantity of 9.92 mg P g-1. The adsorption kinetics and isotherms of PANAF-FeOOH on phosphate were better depicted by pseudo-quadratic kinetics and Langmuir isotherm models, showing a monolayer chemisorption procedure. Inconclusion, this study provides evidence for PANAF-FeOOH as a potential material for the degradation of OP and simultaneous recovery of phosphate. There is potential in the research presented in the article, and it should be published after the following revisions have been made:

Question 1: The abstract would benefit from a clearer statement of the research objectives. While the authors briefly mention that their goal is to enhance the removal of organic phosphorus and phosphate, it would be helpful to state explicitly why this is important (e.g. to prevent water eutrophication and utilize OP resources)

Response: Thanks to your suggestions. We have revised the abstract section. A section on eutrophication in water bodies has been added. The modifications to the paper are as follows:

Phosphorus is one of the important metabolic elements for living organisms, but excess phosphorus in water can lead to eutrophication. At present, the removal of phosphorus in water bodies mainly focuses on inorganic phosphorus, while there is still a lack of research on the removal of organic phosphorus (OP). Therefore, the degradation of OP and synchronous recovery the produced inorganic phosphorus has important significance for the reuse of OP resource and the prevention of water eutrophication.

Question 2: The results section of the abstract could be restructured to make it easier to understand. Consider separating the discussion of OP degradation and phosphate removal into two separate paragraphs, and clearly stating the specific findings for each (e.g. the best FeOOH concentration for OP degradation and phosphate adsorption)

Response: We have considered your suggestion carefully and rewrote the abstract, thank you very much. The rewritten abstract is as follows:

Herein, a novel FeOOH-loaded aminated polyacrylonitrile fiber (PANAF-FeOOH) was constructed to enhance the removal of OP and phosphate. Take phenylphosphonic acid (PPOA) as an example, the results indicated that modification of aminated fiber was beneficial to FeOOH fixation, and the PANAF-FeOOH prepared with 0.3 mol L-1 Fe(OH)3 colloid had the best performance for OP degradation. The PANAF-FeOOH efficiently activated peroxydisulfate (PDS) for the degradation of PPOA with removal efficiency of 99%. Moreover, the PANAF-FeOOH maintained a high removal capacity in five cycles, as well as strong anti-interference in coexisting ion system for OP. In addition, the removal mechanism of PPOA by PANAF-FeOOH is mainly attributed to the enrichment effect PPOA on the fiber surface special microenvironment, which is more conducive to contact with SO4•- and •OH generated by PDS activation. Furthermore, the PANAF-FeOOH prepared with 0.2 mol L-1 Fe(OH)3 colloid possessed excellent phosphate removal capacity with maximal adsorption quantity of 9.92 mg P g-1. The adsorption kinetics and isotherms of PANAF-FeOOH on phosphate were better depicted by pseudo-quadratic kinetics and Langmuir isotherm models, showing a monolayer chemisorption procedure. And the phosphate removal mechanism is mainly due to the strong binding force of iron and the electrostatic force of protonated amine on the PANAF-FeOOH. In conclusion, this study provides evidence for PANAF-FeOOH as a potential material for the degradation of OP and simultaneous recovery of phosphate.

Question 3:  The introduction provides a clear overview of the significance of phosphorus and its pollution problems. However, it would be helpful to provide some quantitative data on the extent of the problem and its impact on the environment and human health.

Response: Thank you for your valuable suggestion. In the introduction, we have added some quantitative data for the eutrophication problem in water bodies. Such as: The high concentration of P (10 mg L-1 ) would cause water eutrophication, lead to the excessive growth of algae and aquatic plants, reduce dissolved oxygen in water and even threaten human safety [3]. Currently, there are still many waters in China that are eutrophic, for example, Chaohu Lake shows total phosphorus concentration averaging 0.215 mg L-1 [4]. And about 65% of European Atlantic coastal waters were eutrophic[5].

Question 4:  Please provide more details on the continuous flow test, such as the flow rate and the concentration of PPOA and phosphate used in the test.

Response: The information about the continuous flow experiment was previously placed in Supporting information. For better understanding by readers, the content has been moved from Supporting information to the main manuscript. The details are as follows:

The continuous flow device is shown in Figure. S3, dried PANAF-FeOOH (300 mg) was placed in a silicone tube (length 100 mm, diameter 5.6 mm), and then injected Chaohu Lake water with an initial concentration of 1 mg P L-1 at a flow rate of 1 mL min-1 with a peristaltic pump (the concentration of Chaohu Lake water was specially adjusted to 1 mg P L-1), the concentration of phosphate in the receiving solution was determined.

Question 5: Please provide more details on the specific procedure and conditions used for the measurement.

Response: Thank you for your suggestion. The specific procedures and measurement conditions are briefly described in sections 2.3 and 2.4 of the manuscript text and in section 3 of the Supporting Information.

Question 6:  Please provide more details on the characterization techniques used to analyze the synthesized materials. For instance, what was the wavelength range used for the FTIR spectra? What was the scanning range for the XRD analysis? What was the step size and scanning rate for the TGA analysis?

Response: The details of the characterization techniques are summarized in section 2 of our support information, as follows:

The surface morphology and element distribution of the fibers were determined by scanning electron microscope (S-4800, Hitachi) and X-ray energy dispersive spectrometer (EDS, X-Max N 150). The changes of functional groups and elemental content (C, N and O) on the fiber surface were measured by Fourier Infrared Spectrometer (Nicolette is50, USA) and elemental analyzer (Vario EL Cube, Germany), respectively. The test method of Fourier Infrared chromatography is to cut fiber into pieces, mix the fiber with potassium bromide, and press them into thin sheets, then measure it in the wavenumber range of 4000-400 cm-1. The internal crystal structure of different fibers is performed by the D/MAX-2500 X-ray diffraction (XRD) (the Confucian Co., Ltd.) within the range of 10°-90° at a speed of 4 ° per minute. Thermal stability of the fibers was tested by a STA409F5TGA/ DSC simultaneous thermal analyzer (Netzsch company, Germany) from room temperature to 800 °C at a rate of 10 °C min-1 in nitrogen atmosphere. The surface structure and elemental composition of the fibers were characterized by X-ray photoelectron spectroscopy (XPS, PERKIN ELMZR), Full-spectrum scan flux energy is 150 eV in 1 eV steps; narrow-spectrum scan flux energy is 50 eV in 0.1 eV steps. The species of free radicals in the reaction system were determined by Electron paramagnetic resonance spectrometer (EPR, Bruker EMXplus) at the 5th min from the beginning of the reaction.

Question 7: In the Results and discussion section, it would be helpful to discuss the practical implications of the results. For example, what are the potential applications of the FeOOH loaded aminated polyacrylonitrile fiber in real-world wastewater treatment? Are there any limitations or challenges that need to be addressed before the material can be used on a larger scale? It would be helpful if the authors provided more information on the potential practical applications of their fiber material. For example, they could discuss how the material could be used in wastewater treatment plants or other industrial processes to remove organic and inorganic phosphorus from wastewater.

Response: Thank you for your valuable suggestions. We have supplemented the application experiments of fibers in real water bodies in section 3.7. the results are as follow:

This study also tested the practical application ability of PANAF-FeOOH to remove phosphorus in actual water and simulated water, respectively. The total phosphorus concentration was adjusted to about 1 mg P L-1 (0.5 mg P L-1 PPOA + 0.5 mg P L-1 KH2PO4) in both the real water (Chaohu Lake, China) and the deionized water. Dried 50 mg of PANAF-FeOOH and 0.5 mmol L-1 of PDS were put into 20 mL of the above water samples, and the remaining total phosphorus concentration was measured after stirring for 1 h, the total phosphorus concentration in the water samples before and after fiber treatment were measured. The results showed that PANAF-FeOOH could effectively remove 83.6% of total phosphorus in the water sample of Chaohu Lake, while in the simulated wastewater, the removal rate reached 90.4%. The application effect of fibers in real water bodies is slightly lower than that of simulated water samples, possibly due to the complex coexisting ion interference in real water bodies. Altogether, the PANAF-FeOOH has excellent application ability for phosphorus removal in real water.

Round 2

Reviewer 3 Report

The paper has been revised.